# Microstructure and Strength of Ti-6Al-4V Samples Additively Manufactured with TiC Heterogeneous Nucleation Site Particles

**DOI:** 10.3390/ma16175974

**Published:** 2023-08-31

**Authors:** Yoshimi Watanabe, Shintaro Yamada, Tadachika Chiba, Hisashi Sato, Seiji Miura, Kenshiro Abe, Tomotsugu Kato

**Affiliations:** 1Department of Physical Science and Engineering, Nagoya Institute of Technology, Nagoya 466-8555, Japan; shintaro.yamada@jp.fujitec.com (S.Y.); tad-chiba@jfe-steel.co.jp (T.C.); sato.hisashi@nitech.ac.jp (H.S.); 2Division of Materials Science and Engineering, Hokkaido University, Sapporo 060-8628, Japan; miura@eng.hokudai.ac.jp; 3Nabtesco Corporation, Tokyo 102-0093, Japan; kenshiro_abe@nabtesco.com (K.A.); tomotsugu_kato@nabtesco.com (T.K.)

**Keywords:** additive manufacturing, powder bed fusion (PBF), Ti-6Al-4V alloy, heterogeneous nucleation site particles, lattice matching, tensile test, single track test

## Abstract

Our research aims to investigate the fabrication of additively manufactured (AMed) Ti-6Al-4V samples under reduced power with the addition of TiC heterogeneous nucleation site particles. For this aim, Ti-6Al-4V samples are fabricated with and without TiC heterogeneous nucleation site particles using an EOS M 290 machine under optimal parameters and reduced power conditions. The microstructure and tensile behavior of the produced samples were studied. In addition, a single-track test was performed to obtain a good understanding of the suppression of gas pores and balling formation with the addition of TiC heterogeneous nucleation site particles. It was found that the formation of gas pores and balling was suppressed with the addition of heterogeneous nucleation site particles within the metallic powder.

## 1. Introduction

Additive manufacturing (AM) has been applied to fabricate various products, including those with a hollow structure, three-dimensional complex porous bodies, and custom-developed products [1]. For the manufacturing of dense metallic parts, powder bed fusion (PBF) and directed energy deposition (DED) are commonly used. All these processes involve the interaction between fed powder and the laser or electron beam that produces the molten pool, leading to rapid melting and solidification [2]. A sharp temperature gradient in the molten pool, due to rapid melting and solidification, usually causes the formation of coarse columnar crystals in most of the metal AM components. In addition, microstructural defects such as pores, un-melted powder, and balling are formed in the AM products, which results in the degradation of the structural integrity and service performance of the products. A significant problem for the metal AM process is identifying how to prevent the occurrence of defects due to the melting and solidification phenomena around the metal pool [3].

It is expected that the shortcomings of metal AM may be overcome with the addition of heterogeneous nucleation site particles in the metallic powder because equiaxed and fine grain structures are promoted during heterogeneous solidification with heterogeneous nucleation site particles [4,5,6,7]. In our previous study [7], it was found that the relative densities of the AMed SUS 316L samples produced with SrO heterogeneous nucleation site particles under medium and low energy density conditions were higher than those produced without SrO. With the addition of a small amount of heterogeneous nucleation site particles, AMed samples with the same density can be produced under a smaller energy density. The effects of TiC heterogeneous nucleation site particles on the formability and microstructure of AMed Ti-6Al-4V samples were also studied with and without TiC heterogeneous nucleation site particles [4,5]. It was found that the addition of TiC heterogeneous nucleation site particles decreased the grain size of the primary β phase in the Ti-6Al-4V samples and increased the formability (decreased the defects) of the samples. However, the enhancing mechanisms for formability in the AMed Ti-6Al-4V samples with the addition of the TiC heterogeneous nucleation site particles are still unclear.

In this study, an investigation of the effectiveness of TiC heterogeneous nucleation site particles for the suppression of gas pores and balling formation in the Ti-6Al-4V alloy fabricated using PBF is carried out. The Ti-6Al-4V samples with and without 0.3 vol%TiC heterogeneous nucleation site particles are fabricated under optimal parameters. Since the PBF process is a layer-by-layer sequential addition process, in which each layer is formed with the combination of single-scan tracks, a single-track test [8,9] was also performed to obtain a good understanding of the suppression of gas pores and balling formation with the addition of TiC heterogeneous nucleation site particles. As will be described later, it was found that the formation of gas pores and balling was suppressed with the addition of heterogeneous nucleation site particles within the metallic powder.

## 2. Lattice Matching between Heterogeneous Nucleation and the Crystallized Phase

First, we study the lattice matching between the TiC heterogeneous nucleation site and crystallized β phase in the Ti-6Al-4V alloy. In this study, the parameter *M* [10,11], which is approximately proportional to the specific misfit strain energy, will be used to discuss the effective heterogeneous nucleation site. The parameter *M* is defined as
*M* = *ε*_x_^2^ + *ε*_y_^2^ + (2/3)*ε*_x_*ε*_y_, (1)
where *ε*_x_ and *ε*_y_ are the principal misfit strains calculated using
*ε*_x_ = (|*x_hetero_*|*a_hetero_* − |*x_cryst_*|*a_cryst_*)/(|*x_cryst_*|*a_cryst_*)(2)
*ε*_y_ = (|*y_hetero_*|*a_hetero_* − |*y_cryst_*|*a_cryst_*)/(|*y_cryst_*|*a_cryst_*)(3)
where *x_hetero_* and *y_hetero_* are principal directions of heterogeneous nucleation site and *x_cryst_* and *y_cryst_* are principal directions of crystallized phase. It has been shown that the epitaxial relationship with a smaller *M* value becomes the preferred relationship. The parameter *M* is also used to predict the effective heterogeneous nucleation site materials, where the favorable heterogeneous nucleation site phase should have a small *M* value between crystallized phases [12]. If the *M* value is less than about 38 × 10^−^^3^, the nucleating agent is potent [13,14].

Exploration of heterogeneous nucleation site particles for the β phase of the Ti-6Al-4V alloy has been performed for some Ti-based intermetallic compositions, as listed in Table 1. The principal misfit strains were calculated along [100]_cryst_//[100]_hetero_ and [010]_cryst_//[010]_hetero_ for the compounds having the CsCl-type structure and along [100]_cryst_//[110]_hetero_ and [010]_cryst_//[−110]_hetero_ for the NaCl-type structure, as this orientation shows better lattice matching. Although it was found that TiZn has a small *M* value with respect to the bcc-β phase of the Ti-6Al-4V alloy (*M* = 2.1 × 10^−^^3^), its melting point is much smaller than the liquidus temperature of the Ti-6Al-4V alloy. Conversely, TiC has a higher melting point, as listed in Table 1. The crystal structure and lattice parameter of TiC are shown in Figure 1a, where the lattice parameter *a* is 0.4333 nm [15]. The (001)_TiC_ plane overlap on //(001)_Ti6Al4V_ is shown in Figure 1b. Using this orientation relationship, the *M* value is calculated to be 8.3 × 10^−^^3^. Moreover, there is only a small density difference between Ti (4.51 Mg/m^3^) and TiC (4.93 Mg/m^3^). Therefore, in this study, TiC particles were used as heterogeneous nucleation site particles.

Since heterogeneous nucleation occurs at the melting point, the *M* value at elevated temperatures must be investigated. Unfortunately, to the best of our knowledge, there is no available data on the temperature dependence of the lattice parameter of the β phase in the Ti-6Al-4V alloy. Temperature dependence of the thermal expansion coefficient for Ti-6Al-4V is reported by Lu et al. to be 8.78 × 10^−^^6^ at 20 °C, 10 × 10^−^^6^ at 205 °C, 11.2 × 10^−^^6^ at 500 °C, 12.3 × 10^−^^6^ at 995 °C, 12.4 × 10^−^^6^ at 1100 °C, 12.42 × 10^−^^6^ at 1200 °C, 12.5 × 10^−^^6^ at 1600 °C, and 12.5 × 10^−^^6^ at 1650 °C [23]. In addition, the lattice parameters for *a*_α-Ti6Al4V_ at room temperature (RT) and *c*_α-Ti6Al4V_ at RT in the α phase at RT are reported to be 0.29216 nm and 0.46699 nm, respectively [24]. Using the Burgers orientation relationship [25]:(4)$$(0001{)}_{\alpha }//(110{)}_{\beta }\;\mathrm{and}\;{\left[11\bar{2}0\right]}_{\alpha }//{\left[-111\right]}_{\beta }$$
the lattice parameter for the β phase at RT, *a*_β-Ti6Al4V_ at RT, can be estimated. However, it has been reported that the distance between the next-nearest atoms in the basal plane of the α phase is larger than that in the β phase [26]; therefore, it cannot be estimated using a simple geometrical approach. In this study, *a*_β-Ti6Al4V_ at RT = 0.324 nm, as estimated using the *a*_β-Ti_/*a*_α-Ti_ ratio of pure Ti at RT, which is 0.3269 [27]/0.295 [26], is used. The temperature dependence of the lattice parameter for the β phase in the Ti-6Al-4V alloy is then estimated, and the results are shown in Figure 2a.

The percentage of linear thermal expansion (Δ*ℓ*) of TiC can be expressed with the following polynomial
(*ℓ*_T_ − *ℓ*_25_)/*ℓ*_25_ = 4.38 × 10^−^^16^ × (*T* + 273)^4^ − 2.48 × 10^−^^12^ × (*T* + 273)^3^ + 5.96 × 10^−^^9^ × (*T* + 273)^2^+ 1.09 × 10^−^^6^ × (*T* + 273) − 7.11 × 10^−^^4^(5)
where *ℓ*_T_ is the length of the sample at the temperature *T* °C [28]. Using this equation, the temperature dependence of the lattice parameter of TiC can be calculated, and the results are also shown in Figure 2a.

As can be seen, the temperature dependence of the lattice parameter of TiC is smaller than that of Ti-6Al-4V. Using the lattice parameters of Ti-6Al-4V and TiC, the *M* values at elevated temperatures are calculated, and the results are shown in Figure 2b. It is seen that the lattice matching is worsened at elevated temperatures because the principal misfit strains have negative values (the atomic space of TiC is smaller than that of Ti-6Al-4V). Nevertheless, the *M* value between TiC and Ti-6Al-4V at 1650 °C is 9.73 × 10^−^^3^, which is much smaller than 38 × 10^−^^3^. Therefore, we can conclude that TiC acts as a good heterogeneous nucleation site for the solidification of Ti-6Al-4V.

## 3. Experimental Procedure

Gas-atomized Ti-6Al-4V powder (45 µm under) and TiC particles (2–5 µm, Kojundo Chemical Laboratory Co., Ltd., Sakado-shi, Japan) were used. The chemical composition of the Ti-6Al-4V powder is listed in Table 2. Mixing of the powders was performed with a 3D motion mixer (Turbula shaker mixer) for 1 h. The container motion during mixing consisted of two rotations of the container around its longitudinal axis and horizontal translation to achieve uniform mixing of the particles [29,30]. The volume fraction of TiC particles in the mixed powder was fixed to 0.3 vol%.

In this study, an EOS M290 machine was used to fabricate three different shaped specimens, i.e., cube-shaped samples with dimensions 10 mm × 10 mm × 10 mm for microstructural observation, cuboid-shaped samples with dimensions 10 mm × 5 mm × 5 mm for density measurement using the X-ray computer tomography (X-CT) method, and samples with dimensions 100 mm × 12 mm × 12 mm for tensile tests, as seen in Figure 3.

Four types of specimens were studied. Two were fabricated under optimal parameters (default EOS M290 scanning parameters) under an energy density of 55.6 J/mm^3^. The samples with and without TiC heterogeneous nucleation site particles fabricated under optimal parameters are referred to as 100% with TiC and 100% w/o TiC, respectively. On the other hand, two types of specimens were fabricated with TiC heterogeneous nucleation site particles under reduced energy density. The reduced energy densities are 50.0 J/mm^3^ and 41.7 J/mm^3^, which are 90% and 75% energy densities of the optimal condition, respectively. The samples fabricated under those conditions are referred to as 90% with TiC and 75% with TiC, respectively.

The density of the as-fabricated cube-shaped specimens was characterized using X-CT (TOSCANER-32300mFD, Toshiba IT & Control System Corporation, Tokyo, Japan). In this study, scans with voxel sizes of 6 μm were performed. Furthermore, the cube-shaped specimens were cut and then mechanically and chemically polished using emery papers, diamond paste containing diamond particles of 1 μm in diameter, and colloidal silica. After that, the polished plane was etched using an etchant (HF:HNO_3_:H_2_O = 1 mL:2 mL:17 mL). Using the polished specimens, the microstructure of the cube-shaped specimens was characterized using an optical microscope (OM, BM51, Olympus). Furthermore, in order to investigate the crystal orientation distribution on the observation plane, the observation plane was electrically polished. The electrical polishing was performed using an electrolyte (C_2_H_6_O:C_2_H_6_O_2_:HClO_4_:H_2_O = 700 mL:100 mL:80 mL:120mL) under a voltage of 40 V for 60 s. Then, the crystal orientation distribution of the observation plane was measured using electron backscattered diffraction (EBSD, DigiView, TSL Solutions, Osaka, Japan). The microstructural observation using EBSD was performed under an acceleration voltage of 15 kV and a step size of 0.2 μm. Subsequently, the crystal orientation distribution obtained using EBSD was analyzed with orientation imaging software (OIM analysis 8.6, TSL Solutions).

Differential thermal analysis (DTA) was carried out with a BRUKER TG-DTA2200SA-HT22. Each powder specimen of approximately 0.02 g was placed in a ZrO_2_-Y_2_O_3_ crucible with an Al_2_O_3_ lid and heated and then cooled at a rate of 0.17 °C/s under a flowing Ar atmosphere. Occurrence of events during heating and cooling were detected with characteristic peaks on a DTA chart; the events in each alloy in the present study could be identified as a solidus temperature and a liquidus temperature by combining the previous report [31] and ThermoCalc calculation [4]. There are many small fluctuations in the curves, which may be due to difficulties in temperature control in the temperature range of about 250 °C higher than the melting point of Ni. Also, the newly adopted ZrO_2_-Y_2_O_3_ crucibles might be one of the reasons for the fluctuation by changing heat flow in the furnace. The melting points of Ni and Au were also measured occasionally for calibration.

Dogbone samples were cut from the block using wire electrical discharge machining (EDM), following the dimensions outlined in Figure 4. The tensile tests were carried out using an Instron 5982 Universal Testing System with the load perpendicular to the direction. Tensile tests were conducted at RT at a crosshead speed of 0.45 mm/min. Two tensile tests were carried out for each type of specimen. Tensile strength, 0.2% proof stress, and total elongation of the samples were studied. Fracture surfaces for tensile testing were studied using scanning electron microscopy (SEM) fractographs.

To study the effects of TiC heterogeneous nucleation site particles on the formation of gas pores and balling, a single-track test [8,9] was also carried out. Two types of samples were used. Single-track laser melting experiments were carried out on a single layer of powder bed directly fabricated on the base plate (Figure 5a) and on a pre-AMed plate with dimensions 17 mm × 34 mm × 5 mm (Figure 5b). The former and later experiments are referred to as the single-track test w/o pre-AMed plate and the single-track test with pre-AMed plate, respectively. For each specimen, a single pass of a laser beam along a 10 mm × 10 mm square was irradiated. The single-track tests were carried out under two different laser powers, i.e., 37.8 J/mm^3^ (68% for optimal laser power) and 13.9 J/mm^3^ (25% for optimal laser power). Hereafter, the former and later tests are referred to as the 68% power single-track test and the 25% power single-track test, respectively. The sample notation is summarized in Table 3. After the single-track test, each track was observed using a digital microscope camera (VK-X1100, KEYENCE, Osaka, Japan) to observe the formation behavior of the balling.

## 4. Results and Discussion

### 4.1. Powder

Figure 6a–c shows the scanning electron microscope (SEM) images of the gas-atomized Ti-6Al-4V powder, TiC heterogeneous nucleation site particles, and Ti-6Al-4V powder with 0.3 vol%TiC heterogeneous nucleation site, respectively. The Ti-6Al-4V powder has a spherical morphology, and only a few satellites were observed (Figure 6a), while the TiC particles have irregular shapes (Figure 6b). As seen in Figure 6c, the TiC particles are located on the surface of the Ti-6Al-4V powder.

Figure 7 shows the DTA curve. The Ti-6Al-4V alloy powder without TiC particles seems to start melting at 1700 °C during heating, and then solidification occurs at 1700 °C during cooling. This melting point is almost consistent with the result reported by Mizukami et al. (1700 °C) [31]. On the other hand, the Ti-6Al-4V alloy powder with TiC particles seems to start to melt at 1655 °C during heating, and almost corresponding reactions start at 1700 °C during cooling. The difference in the melting point between the Ti-6Al-4V alloy powders with and without TiC particles of about 35 °C is consistent with the difference between the melting point of pure Ti and the eutectic temperature of the Ti-TiC reaction of about 20 °C. This may suggest that carbon in the TiC particles diffused into and reacted with neighboring Ti-6Al-4V alloy [32] powder during the heating process in DTA before reaching the eutectic melting temperature since the heating rate is very slow (0.17 °C/s), resulting in the occurrence of eutectic melting at 1655 °C during heating.

### 4.2. Microstructure of Fabricated Samples

The etched cross-sections of 100% w/o TiC sample, 100% with TiC sample, 90% with TiC sample, and 75% with TiC sample were observed using OM to study the grain structure. The results are shown in Figure 8a–d, respectively. The columnar grains aligned along the building direction are observed for all samples, including the samples fabricated with TiC heterogeneous nucleation site particles. The TiC heterogeneous nucleation site particles cannot suppress the formation of columnar grains in the AMed Ti-6Al-4V samples. The high magnification cross-sections of 100% with TiC sample and 75% with TiC sample are shown in Figure 8e and Figure 8f, respectively. TiC particles can be observed, which act as the heterogeneous nucleation site particle for the solidification of Ti-6Al-4V. This is in contrast to the DTA results, by which the equilibrium state can be achieved.

To observe pore formation, un-etched cross-sections of the samples were observed using OM. A typical microstructure of 100% w/o TiC sample, 100% with TiC sample, 90% with TiC sample, and 75% with TiC sample observed using OM are shown in Figure 9a–d, respectively. Although the 100% w/o TiC sample contains a few pores, as shown in Figure 9a, the samples with TiC particles were almost fully dense regardless of the energy density, as shown in Figure 9b–d. The relative density of the samples characterized using X-CT data is shown in Table 4. It is found that the relative density of the sample can be improved with the addition of TiC heterogeneous nucleation site particles. These are in agreement with previous results of the studies [4,5]. It is worthwhile to notice that the relative density of the 75% with TiC sample is larger than that of the 100% w/o TiC sample. Therefore, the addition of TiC heterogeneous nucleation site particles can achieve an eco-AM process under reduced input energy.

Figure 10a,b shows inverse pole figure (IPF) and phase maps of the cube-shaped specimens without and with TiC particles, respectively. In these IPF maps, the vertical direction corresponds to the building direction. The laser power of both specimens is 100%. As shown in Figure 10, both specimens have a typical microstructure of Ti-6Al-4V alloy, which has mainly the α phase. Moreover, the microstructures of both specimens have columnar structure of the prior β grain regardless of the addition of TiC particles.

In order to observe the size of the prior β grains, reconstructed images of the prior β grains were made for both specimens. The reconstruction from the α phase to the β phase is performed based on the crystal orientation relationship in Equation (4) using orientation imaging software (OIM analysis 8.6, TSL Solutions). The reconstruction parameters used in this study were a tolerance angle of 2.0° and a minimum probability ratio of 1.15. Figure 11a and Figure 11b show IPF maps for the prior β grains reconstructed from Figure 10a and Figure 10b, respectively. Although the grain size of the prior β grains cannot be measured due to their large grain size, it is seen that the specimen with TiC particles contains relatively small grains. Hence, the addition of TiC particles would reduce the size of the prior β grains. This is in good agreement with the results of the previous study [4].

Figure 12a and Figure 12b show the IPF of the prior β grains in the cube-shaped specimen without and with tTiC particles, respectively. These IPFs present crystal plane orientation distributions observed from the building direction. As seen in Figure 12a, the <001> and <101> directions of the specimen without TiC particles are aligned to the building direction. Since the <001> direction is the easy growth direction of bcc metals, this crystallographic texture is formed by solidification [33]. On the other hand, the crystal orientation of the specimen with TiC particles is broadly distributed, although the <111> direction of this specimen is aligned with the building direction. This suggests that TiC particles act as the heterogeneous nucleation site during 3D printing. Therefore, the addition of TiC particles is effective in reducing the typical solidification texture of the AMed specimen.

### 4.3. Tensile Test

A summary of the tensile behaviors of the samples is shown in Figure 13. For both the tensile strength (1031 MPa for the 100% w/o TiC sample and 1081 MPa for the 100% with TiC sample) and the 0.2% proof stress (970 MPa and 1011 MPa, respectively) values are about 4 pct higher in the 100% with TiC sample. The higher tensile strength and 0.2% proof stress of the sample fabricated with TiC particles are attributed to the formation of a smaller grain size as well as a smaller number of pores. It is also found that the tensile strength and 0.2% proof stress of the samples fabricated with TiC particles under reduced input power are better than those of the 100% w/o TiC sample.

In addition, comparing the 100% w/o TiC sample and the 100%, 90%, and 75% with TiC samples, the samples fabricated with TiC particles have higher total elongation than those of the samples fabricated without TiC particles regardless of the input power. This is because of the dense structure obtained for the samples fabricated with TiC particles compared with that of the sample fabricated without TiC particles.

Fractographs of the 100% w/o TiC sample, the 100% with TiC sample, the 90% with TiC, and the 75% with TiC sample after tensile tests are shown in Figure 14a–d, respectively, where Figure 14(a-1–d-1,a-2–d-2) show the low and high magnification features, respectively. Notable differences among Figure 14(a-1,d-1) cannot be observed. It is seen from the high magnification photographs shown in Figure 14(a-2–d-2) that only a ductile failure and typical dimple failure morphology [34] are observed for all samples. Although the crack initiation usually occurs at a non-metallic inclusion, no TiC particles are observed on the fracture surface. Therefore, we can conclude that the negative effects of TiC particles on fracture are small.

### 4.4. Single Track Test

Figure 15 shows the macro-level morphologies of the 68% power single tracks w/o pre-AMed plate observed using a digital microscope camera. The used powders shown in Figure 15a and Figure 15b are the Ti-6Al-4V powder without and with TiC particles, respectively. Height-based coloring is shown in the lower figures. Very few metal droplets were observed for both tracks formed using the Ti-6Al-4V powder without and with TiC particles. In addition, stable tracks were fabricated for both the Ti-6Al-4V powder without and with TiC particles, which is clearly visible in the height-based coloring. The molten metal spread steadily across the base plate, forming a continuous track with good physical bonding between the consolidated track and the base plate.

The macro-level morphologies of the 25% power single tracks w/o pre-AMed plate are shown in Figure 16. Figure 16a and Figure 16b show the 25% power single tracks using the Ti-6Al-4V powder without and with TiC particles, respectively. Again, a notable difference was not observed between Figure 16a and Figure 16b, and very few metal droplets were observed for both tracks fabricated using the Ti-6Al-4V powder regardless of being without and with TiC particles. However, discontinuous tracks can be observed for the 25% single tracks since the input energy was not sufficient to melt both the particles and the base plate. This can be seen clearly by comparing the macro-level photographs in Figure 15a and Figure 16a, as well as Figure 15b and Figure 16b. Therefore, drawing a continuous track requires a certain energy density, which may be achieved with the default scanning parameters.

Although very few metal droplets were formed with the single-track tests w/o pre-AMed plate regardless the input power and addition of TiC particles, a balling morphology was formed in the single-track tests with pre-AMed plate. The macro-level morphologies of the 68% power single tracks with pre-AMed plate using the Ti-6Al-4V powder without and with TiC particles are shown in Figure 17a and Figure 17b, respectively. It is worth noting that many metal droplets were found in these samples, although continuous tracks can be observed. Only the difference in the test conditions between Figure 16 and Figure 17 is the substrate; the surface conditions of the substrate strongly influence the formability of additive manufacturing. Similar observations were found for the 25% power single tracks with the pre-AMed plate. Figure 17c and Figure 17d show the 25% power single tracks using the Ti-6Al-4V powder without and with TiC particles, respectively.

It is reported that this mechanism may be associated with the insufficient melting of the substrate material and its poor wetting [9]. The substrate remained non-melted, while the melted particles preferred to coalesce into tiny droplets and then rapidly consolidate into individual balls. The balling effect in the laser beam PBF can also be explained by the surface tension, which breaks along the melt track into isolated droplets to minimize the surface area [35]. To study the balling morphology quantitatively, the number of metal droplets formed in the single tracks with the pre-AMed plate was counted, and the results are presented in Figure 18a. In this figure, the left and right groups are the results for the 68% power and 25% power single tracks with the pre-AMed plate, respectively, and the left and right bars in each group are fabricated using the Ti-6Al-4V powder without and with TiC particles, respectively. This figure clearly indicates that a larger number of metal droplets is observed in the samples fabricated with TiC heterogeneous nucleation site particles regardless of the input power, contrary to expectations. However, one may notice in Figure 17 that the metal droplets found in the single track fabricated with TiC particles are smaller than those fabricated without TiC particles. Therefore, the diameter of metal droplets formed in the 68% power single tracks with the pre-AMed plate was measured, and average droplet size results are shown in Figure 18b. It is seen that smaller average droplet size is found for the single track fabricated with TiC particles. Therefore, the total volume of metal droplets formed with TiC particles is smaller than that without TiC particles, as shown in Figure 18c. One of the possible reasons for better formability with the addition of TiC particles during the additive manufacture of a Ti-6Al-4V sample is not by suppression of metal droplet formation but by decreasing the size of metal droplets.

It is also reported that the balling can be divided into two types: ellipsoidal balls, which are caused by worsened wetting ability and are detrimental to AM fabrication quality with a dimension of about 500 μm, and spherical balls, which have no obvious detriment to AM quality with a dimension of about 10 μm [36]. Since the diameter of the observed metal droplets is about 20 μm, the formed metal droplets may have no obvious detriment to AM quality.

## 5. Conclusions

In this study, the effects of TiC heterogeneous nucleation site particles on the microstructure and strength of Ti-6Al-4V samples fabricated using powder bed fusion (PBF) are studied. The results can be summarized as follows.
(1)The formation of pores can be reduced, and the relative density of the sample can be improved with the addition of TiC particles. Thus, the formability of AM can be improved with the addition of heterogeneous nucleation site particles.(2)A higher tensile strength, 0.2% proof stress, and total elongation of the sample fabricated with TiC particles were found. Moreover, the negative effects of TiC particles on fracture were small.(3)The surface conditions of the substrate were found to be strongly influenced by the formability of additive manufacturing.(4)One of the possible reasons for better formability with the addition of TiC particles during the additive manufacture of a Ti-6Al-4V sample is not by suppression of metal droplet formation but by decreasing the size of metal droplets.(5)The addition of TiC heterogeneous nucleation site particles can achieve an eco-AM process since the sample with the same quality can be fabricated under reduced input energy.

## Figures and Tables

**Figure 1 materials-16-05974-f001:**
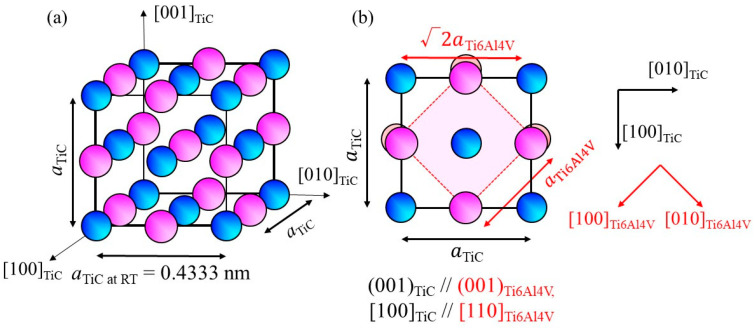
(**a**) Crystal structure of TiC and (**b**) atomic arrangement of the (001)_TiC_ plane superimposed on the atomic arrangement of (001)_Ti6Al4V_.

**Figure 2 materials-16-05974-f002:**
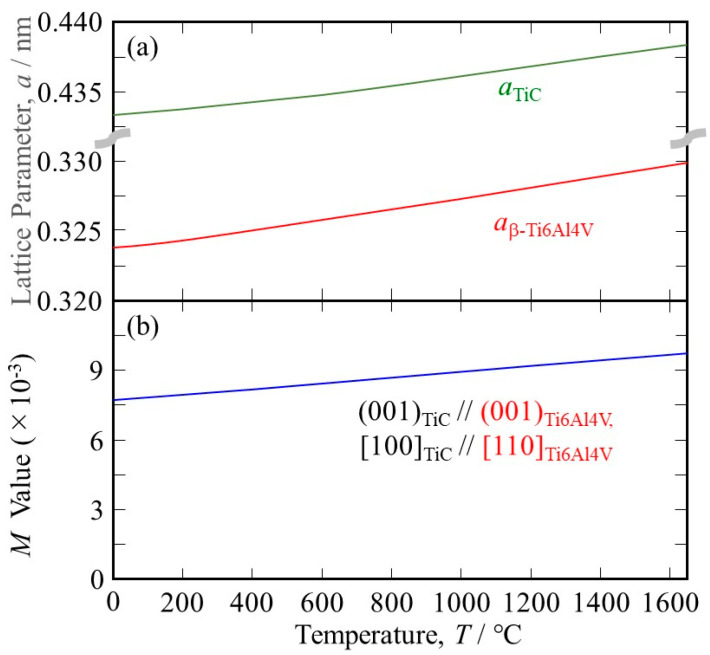
(**a**) The temperature dependence of (**a**) the lattice parameters of TiC and the β phase in the Ti-6Al-4V alloy and (**b**) the *M* values between the TiC heterogeneous nucleation site and the β phase in the Ti-6Al-4V alloy under (001)_TiC_//(001)_Ti6Al4V_, [100]_TiC_//[110]_Ti6Al4V_.

**Figure 3 materials-16-05974-f003:**
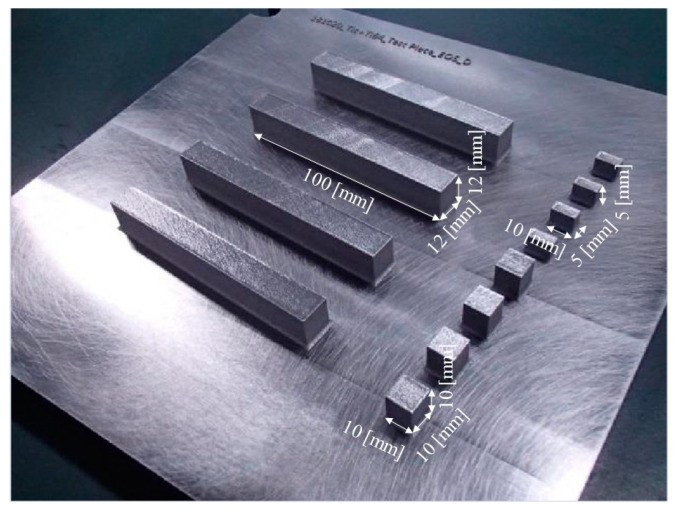
Photograph showing cube-shaped samples with dimensions 10 mm × 10 mm × 10 mm for microstructural observation, cuboid-shaped samples with dimensions 10 mm × 5 mm × 5 mm for density measurement using the X-CT method, and samples with dimensions 100 mm × 12 mm × 12 mm for tensile tests.

**Figure 4 materials-16-05974-f004:**
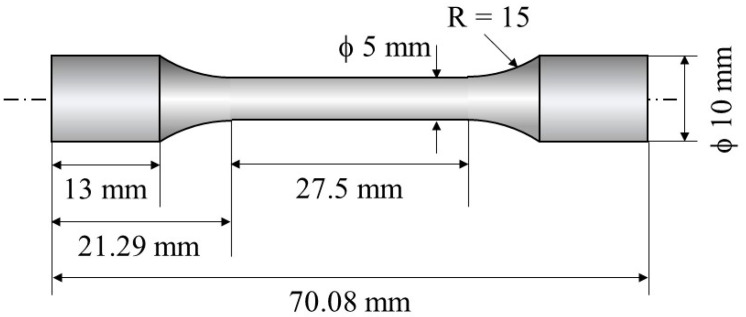
Schematic illustration showing a dogbone sample used for tensile tests.

**Figure 5 materials-16-05974-f005:**
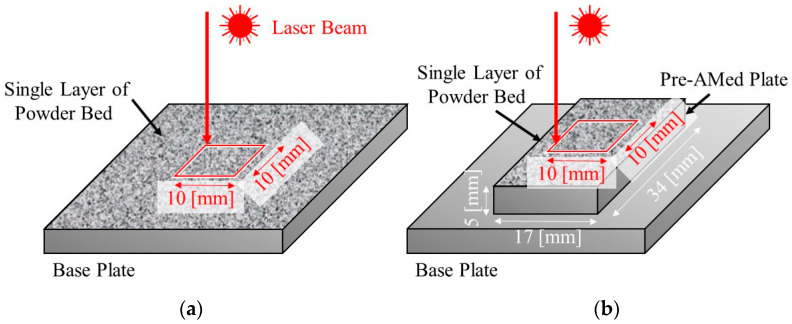
Schematic illustrations showing the single-track laser melting tests. Single passes of a laser beam irradiation along a 10 mm × 10 mm square were carried out on (**a**) a powder bed fabricated on the base plate (single-track test w/o pre-AMed plate) and (**b**) on a pre-AMed plate with dimensions 17 mm × 34 mm × 5 mm (single-track test with pre-AMed plate).

**Figure 6 materials-16-05974-f006:**
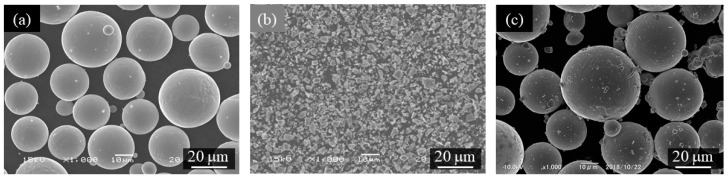
The scanning electron microscope (SEM) images of (**a**) gas-atomized Ti-6Al-4V powder, (**b**) TiC heterogeneous nucleation site particles, and (**c**) Ti-6Al-4V powder with 0.3 vol%TiC heterogeneous nucleation site.

**Figure 7 materials-16-05974-f007:**
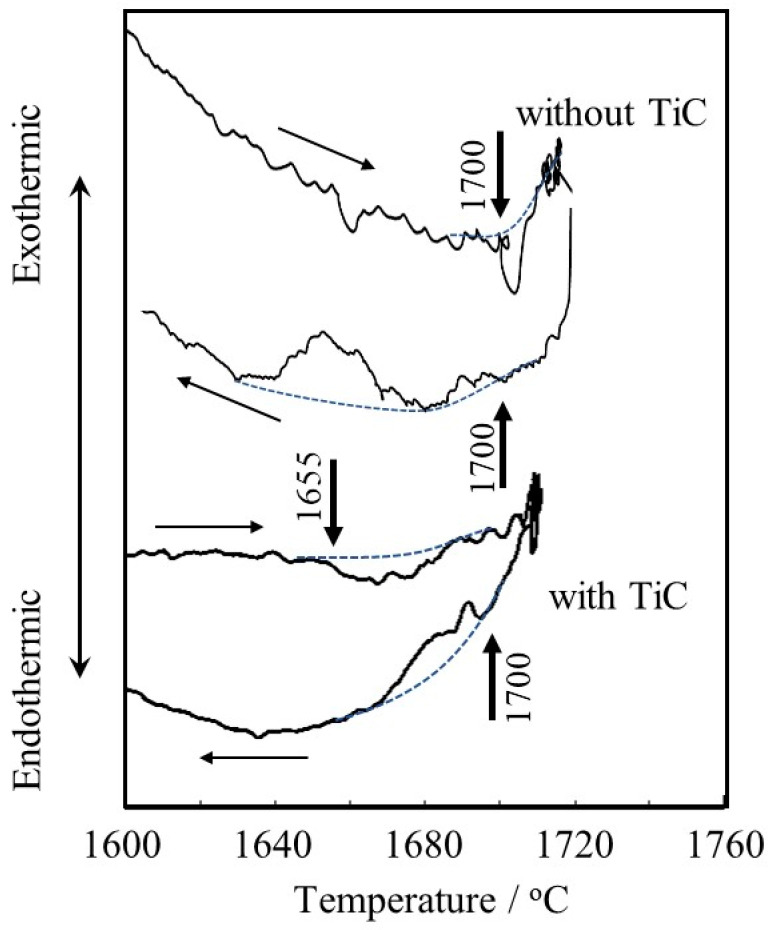
DTA curves of Ti-6Al-4V alloy powders with and without TiC particles. Dotted curves are baselines and arrows indicate the start points of melting (1700 °C for the powder w/o TiC and 1655 °C for the powder with TiC) or solidification (1700 °C for the powder w/o TiC and 1700 °C for the powder with TiC), respectively.

**Figure 8 materials-16-05974-f008:**
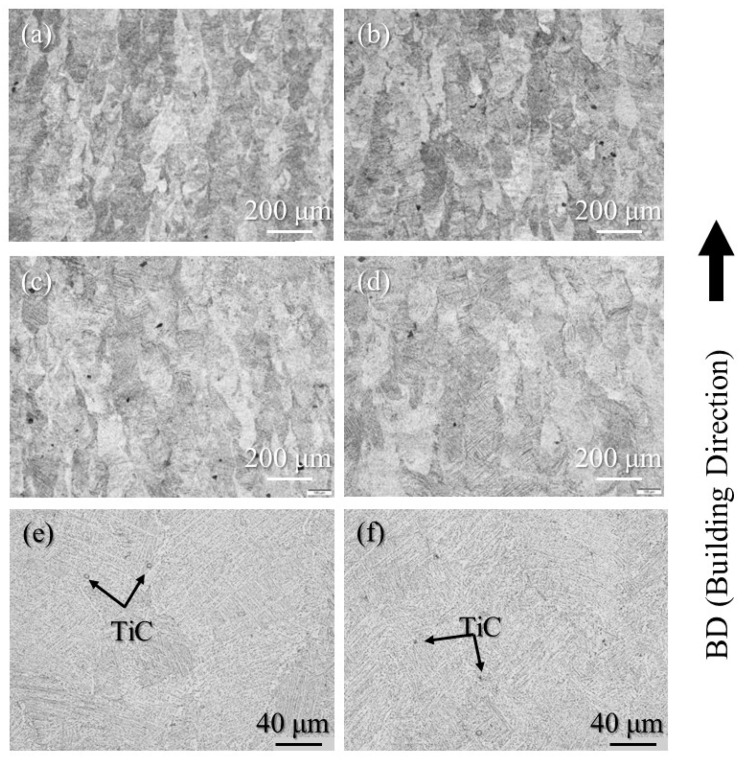
The etched cross-sections of the (**a**) 100% w/o TiC sample, (**b**) 100% with TiC sample, (**c**) 90% with TiC sample, and (**d**) 75% with TiC sample observed using OM (low magnification). High magnification cross-sections of the (**e**) 100% with TiC sample and (**f**) 75% with TiC sample.

**Figure 9 materials-16-05974-f009:**
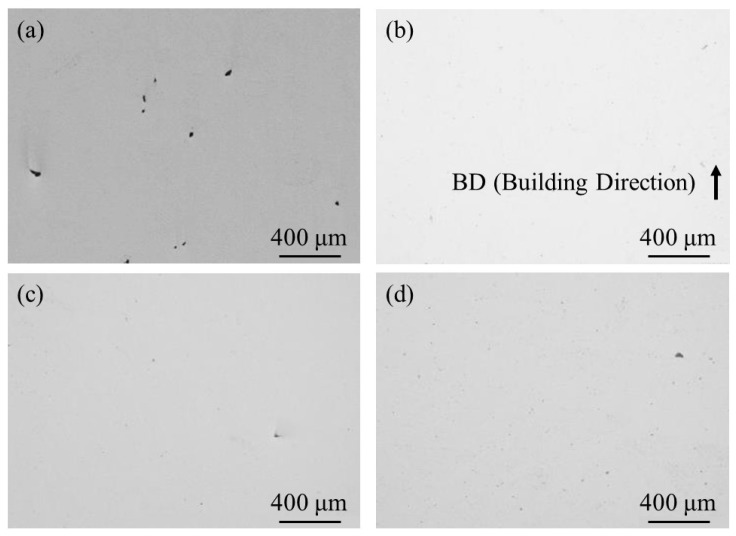
A typical microstructure of un-etched cross-sections of the (**a**) 100% w/o TiC sample, (**b**) 100% with TiC sample, (**c**) 90% with TiC sample, and (**d**) 75% with TiC sample observed using OM.

**Figure 10 materials-16-05974-f010:**
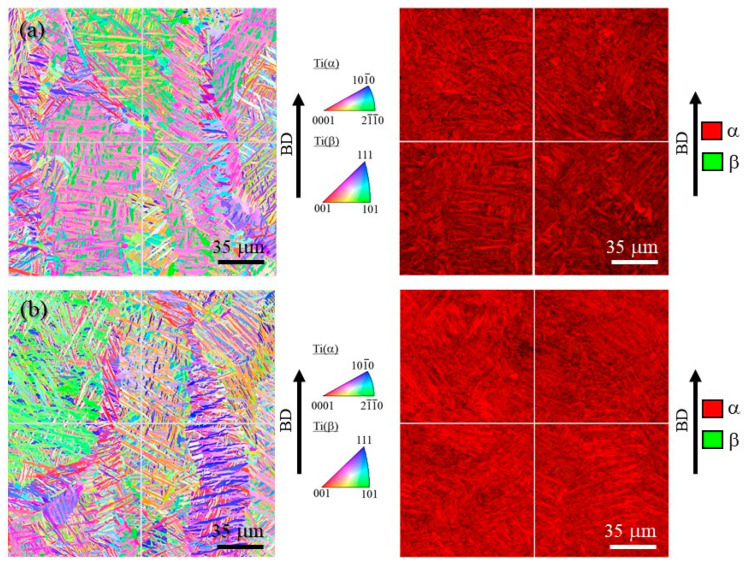
IPF map (**left**) and phase map (**right**) of the cube-shaped specimens (**a**) without and (**b**) with TiC particles. The vertical direction is the building direction.

**Figure 11 materials-16-05974-f011:**
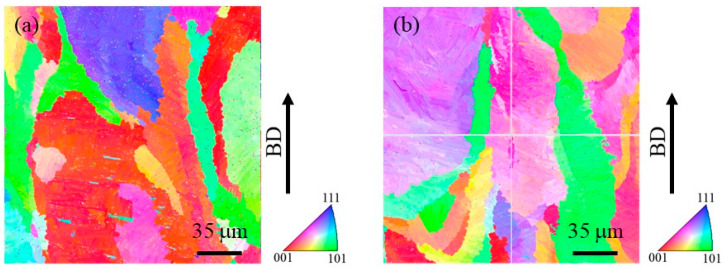
Reconstructed IPF maps for the prior β grains in the cube-shaped specimens (**a**) without and (**b**) with TiC particles. The vertical direction is the building direction. These images were obtained by reconstruction using Figure 10.

**Figure 12 materials-16-05974-f012:**
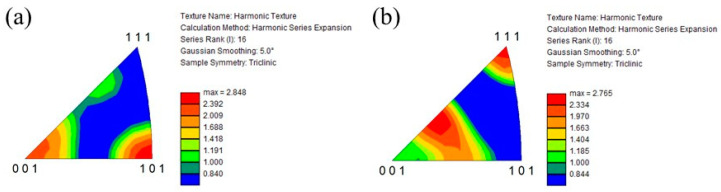
IPF of the prior β grains in the cube-shaped specimens (**a**) without and (**b**) with TiC particles. These crystal plane directions are plane directions observed from the building direction.

**Figure 13 materials-16-05974-f013:**
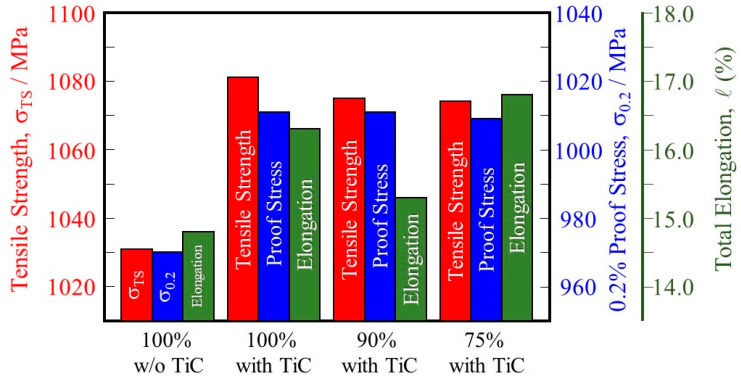
Tensile strength, 0.2% proof stress, and total elongation of the 100% w/o TiC sample, the 100% with TiC sample, the 90% with TiC, and the 75% with TiC sample.

**Figure 14 materials-16-05974-f014:**
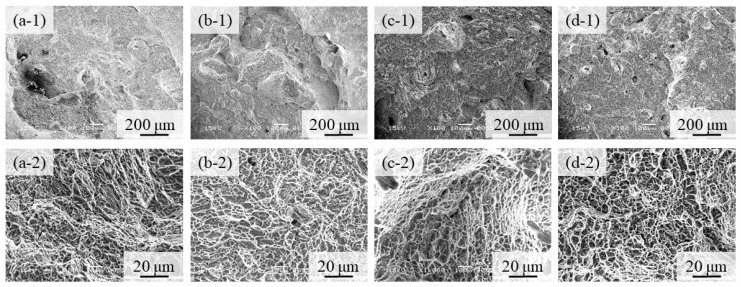
Fractographs of (**a**) the 100% w/o TiC sample, (**b**) the 100% with TiC sample, (**c**) the 90% with TiC, and (**d**) the 75% with TiC sample after tensile tests. Low and high magnification features are shown in (**a-1**–**d-1**) and (**a-2**–**d-2**), respectively.

**Figure 15 materials-16-05974-f015:**
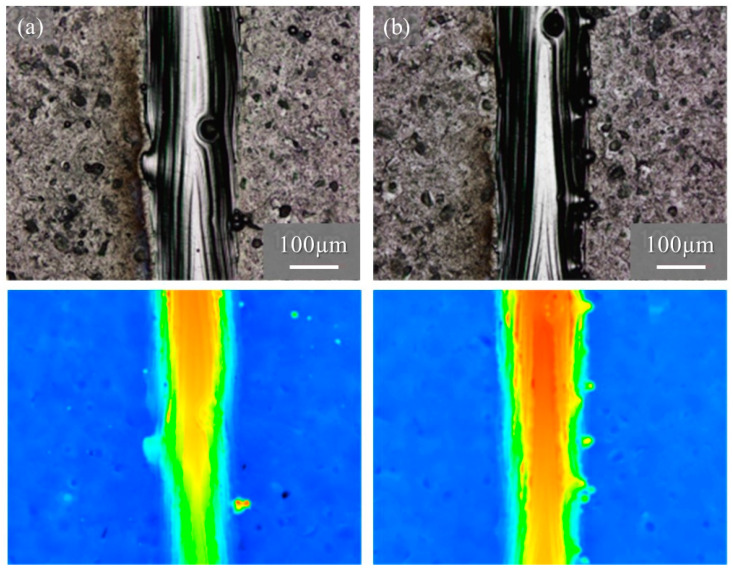
The macro-level morphologies of the 68% power single tracks w/o pre-AMed plate using (**a**) Ti-6Al-4V powder without TiC particles and (**b**) with TiC particles. Height-based coloring is shown in the lower figures. Height increases as the color code changes from blue to red.

**Figure 16 materials-16-05974-f016:**
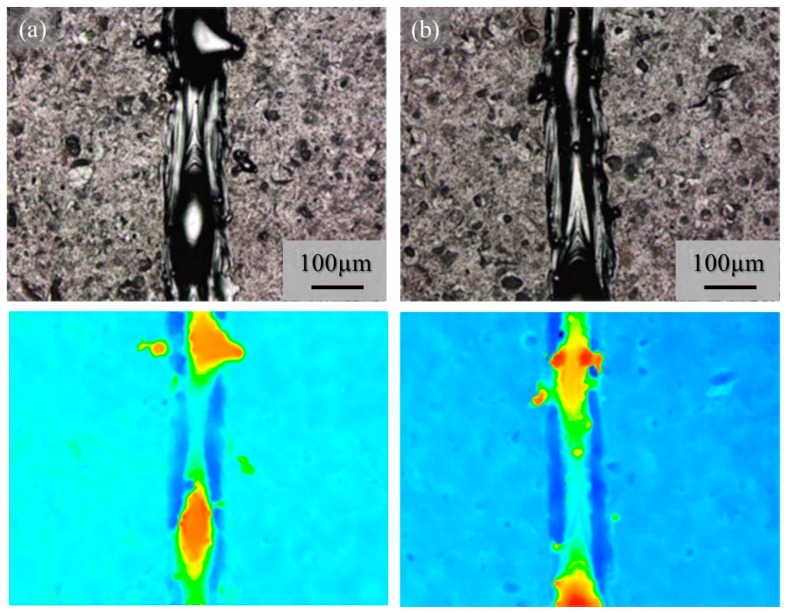
The macro-level morphologies of the 25% power single tracks w/o pre-AMed plate using (**a**) Ti-6Al-4V powder without TiC particles and (**b**) with TiC particles. Height-based coloring is shown in the lower figures. Height increases as the color code changes from blue to red.

**Figure 17 materials-16-05974-f017:**
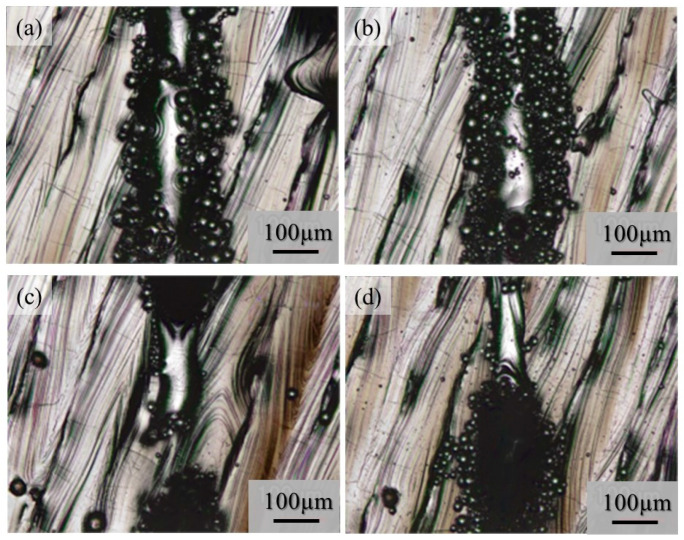
The macro-level morphologies of the 68% power (**a**,**b**) and 25% power (**c**,**d**) single tracks with a pre-AMed plate using the Ti-6Al-4V powder without TiC particles (**a**,**c**) and with TiC particles (**b**,**d**).

**Figure 18 materials-16-05974-f018:**
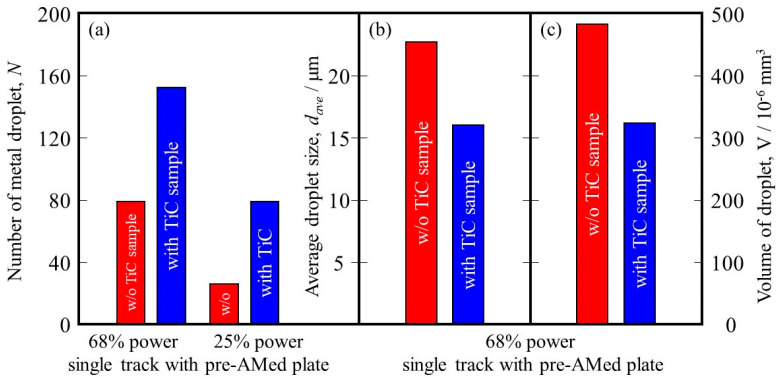
(**a**) The number of metal droplets observed using the single-track tests with a pre-AMed plate. (**b**) Average size and (**c**) volume of metal droplets formed in the 68% power single tracks. The left bar and right bar are the data from the w/o TiC sample and the with TiC sample, respectively.

**Table 1 materials-16-05974-t001:** Crystal structure, melting point, lattice parameter, and parameter *M* of some Ti-based compounds.

	Crystal Structure	Melting Point, *T_M_*/°C	Lattice Parameter at RT, *a*/nm	Parameter *M*
TiFe	CsCl	1317 [16]	0.298 [17]	17.7 × 10^−3^
TiNi	CsCl	1310 [16]	0.3015 [18]	12.9 × 10^−3^
TiCo	CsCl	1325 [16]	0.2995 [19]	15.2 × 10^−3^
TiZn	CsCl	~950 [20]	0.315 [21]	2.1 × 10^−3^
TiC	NaCl	3067 [16]	0.4333 [15]	8.3 × 10^−3^
TiN	NaCl	3290 [16]	0.4241 [22]	14.8 × 10^−3^
Ti-6Al-4V	bcc/hcp	1650 [16]	0.324	

**Table 2 materials-16-05974-t002:** The chemical composition (mass%) of the used Ti-6Al-4V powder.

Al	V	C	O	N	H	Fe	Si
6.44	4.02	0.015	0.11	0.02	0.0024	0.19	0.013

**Table 3 materials-16-05974-t003:** Sample notations of single-track tests.

Energy Density	Pre-AMed Plate	
	w/o Pre-AMed Plate	with Pre-AMed Plate
37.8 J/mm^3^	68% power single track w/o pre-AMed plate	68% power single track with pre-AMed plate
13.9 J/mm^3^	25% power single track w/o pre-AMed plate	25% power single track with pre-AMed plate

**Table 4 materials-16-05974-t004:** The relative density of the AMed samples characterized using X-CT data.

Sample	100% w/o TiC	100% with TiC	90% with TiC	75% with TiC
Relative density	99.96	99.98	99.99	99.98

## Data Availability

Not applicable.

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
