# Peer review of "Microstructure and Strength of Ti-6Al-4V Samples Additively Manufactured with TiC Heterogeneous Nucleation Site Particles"

_materials, 2023, doi:10.3390/ma16175974_

Round 1

Reviewer 1 Report

First of all, I would like to thank you very much for choosing our journal for your article. It is a comprehensive and detailed work. I would like to read the article again after answering the questions I asked.

Can you provide more information on how the xhetero, yhetero, xcryst, and ycryst values were obtained? Specifically, can you elaborate on the techniques used to determine these principal directions?

You mentioned that the M value is less than about 38 × 10^-3 for a potent nucleating agent. Is this an empirically determined threshold or derived from theoretical considerations?

When you calculate the M value for the lattice matching between TiC and Ti-6Al-4V, how do you handle the uncertainty inherent in the measurements, and how does it affect the accuracy of the M value calculation?

You mentioned that TiC has been selected as the heterogeneous nucleation site particles for AM of Ti-6Al-4V in your previous studies. Could you expand on why you chose TiC specifically?

You have used the thermal expansion coefficient of Ti-6Al-4V reported by Lu et al., how does this value compare with other studies? Are there any discrepancies in the literature regarding this parameter?

In your calculations, you've assumed the percentage of linear thermal expansion (Δλ) of TiC as a polynomial function. Could you explain the rationale behind this choice and are there any limitations associated with it?

How is the temperature dependence of the lattice parameter of TiC and Ti-6Al-4V different and what implications does this difference have for their interaction?

Could you clarify why the principal misfit strains have negative values at elevated temperatures, and how does it influence the lattice matching?

It's observed that the lattice matching is worsened at elevated temperature, yet the M value at 1650°C is still much smaller than 38 × 10^-3. Could you discuss any potential impact of this worsened lattice matching on the performance of the material?

Could you explain how you determined the relative density of the samples using X-CT data?

Your study concludes that the addition of TiC heterogeneous nucleation site particles improves the relative density of the sample. Does this result match your initial expectations?

It seems you have several different compositions of samples with varying percentages of TiC. How were these specific percentages chosen and do you anticipate different outcomes with different proportions?

Could you clarify why the relative density of a 75% TiC sample is larger than that of a 100% w/o TiC sample?

You suggest that the addition of TiC heterogeneous nucleation site particles can achieve an eco-AM process under reduced input energy. Could you elaborate on this conclusion? What exactly do you mean by an "eco-AM process"?

It is mentioned that the grain size of the prior β grains cannot be measured due to its large size. Does this limitation affect the conclusions of your study and are there any alternative methods you could use to determine the grain size?

Your study concludes that the addition of TiC particles reduces the size of prior β grains. Can you explain why this happens and what are the implications of having smaller grains in the final material?

How do the crystal plane orientation distributions differ between the specimen without TiC particles and the specimen with TiC particles? Can you elaborate on what you believe is causing these differences?

You mention that <001> direction is easy growth direction of bcc metals, could you elaborate on this concept and its implications on your results?

How might the presence of the TiC particles influence the physical bonding between the consolidated track and the base plate? Have you observed any alterations in bonding strength or quality?

Could you elaborate on why the tracks were discontinuous at 25% power? Are there any particular mechanisms at play due to the reduced energy input?

Based on your observations, what would you suggest as the minimum energy density required for the formation of a continuous track?

Can you explain why there was no notable difference observed between the Ti-6Al-4V powder without and with TiC particles?

In your experiment, have you noticed any effect of TiC particles on the thermal dynamics or heat distribution during the additive manufacturing process?

Author Response

We wish to express our appreciation for the comments and suggestions from reviewer. Our incorporation of reviewer’s comments is attached. Please see the attachment.  

Reviewer 2 Report

The authors presented a microstructural study on Additively Manufactured Ti-6Al-4V  with TiC Heterogeneous Nucleation Site Particles and demonstrated that it has the higher density (less porosity) and strength than the corresponding material without TiC. The information is valuable.

One question is on the DSC curve, whether the authors could point out all the endothermic and exothermic reactions on the DSC curves of Ti64 w/o TiC, as the material going through several phase transformations? 

There are also some minor English typos, e.g., on p. 1-2, "it is found..." should be "it was found" followed by past tense.

There are also some minor English typos, e.g., on p. 1-2, "it is found..." should be "it was found" followed by past tense.

Author Response

(The authors gave the same response as above.)
